# Deep Insight on Land Use/Land Cover Geospatial Assessment through Internet-Based Validation Tool in Upper Karkheh River Basin (KRB), South-West Iran

Sina Mallah [1,2], Manouchehr Gorji [1,*], Mohammad Reza Balali [3], Hossein Asadi [1], Naser Davatgar [2], Hojjat Varmazyari [4], Anna Maria Stellacci [5,*] and Mirko Castellini [6]

1   Department of Soil Science Engineering, University of Tehran, Karaj 77871-31587, Iran; s.mallah@ut.ac.ir (S.M.); hoasadi@ut.ac.ir (H.A.)
2   Department of Soil Physics and Irrigation, Soil and Water Research Institute, Agricultural Research, Education and Extension Organization (AREEO), Karaj 31779-93545, Iran
3   Department of Soil Chemistry, Fertility and Plant Nutrition, Soil and Water Research Institute, Agricultural Research, Education and Extension Organization (AREEO), Karaj 31779-93545, Iran; m.balali@areeo.ac.ir
4   Department of Agricultural Management and Development, University of Tehran, Karaj 31779-93545, Iran
5   Department of Soil, Plant and Food Sciences, University of Bari Aldo Moro, Via Amendola 165/A, 70126 Bari, Italy
6   Council for Agricultural Research and Economics-Research Center for Agriculture and Environment (CREA-AA), Via C. Ulpiani 5, 70125 Bari, Italy
*   Correspondence: mgorji@ut.ac.ir (M.G.); annamaria.stellacci@uniba.it (A.M.S.)

**Abstract:** Recently, the demand for high-quality land use/land cover (LULC) information for near-real-time crop type mapping, in particular for multi-relief landscapes, has increased. While the LULC classes are inherently imbalanced, the statistics generally overestimate the majority classes and underestimate the minority ones. Therefore, the aim of this study was to assess the classes of the 10 m European Satellite Agency (ESA) WorldCover 2020 land use/land cover product with the support of the Google Earth Engine (GEE) in the Honam sub-basin, south-west Iran, using the LACOVAL (validation tool for regional-scale land cover and land cover change) online platform. The effect of imbalanced ground truth has also been explored. Four sampling schemes were employed on a total of 720 collected ground truth points over approximately 14,100 ha. The grassland and cropland totally canopied 94% of the study area, while barren land, shrubland, trees and built-up covered the rest. The results of the validation accuracy showed that the equalized sampling scheme was more realistically successful than the others in terms of roughly the same overall accuracy (91.6%), mean user's accuracy (91.6%), mean producers' accuracy (91.9%), mean partial portmanteau (91.9%) and kappa (0.9). The product was statistically improved to 93.5% $\pm$ 0.04 by the assembling approach and segmented with the help of supplementary datasets and visual interpretation. The findings confirmed that, in mapping LULC, data of classes should be balanced before accuracy assessment. It is concluded that the product is a reliable dataset for environmental modeling at the regional scale but needs some modifications for barren land and grassland classes in mountainous semi-arid regions of the globe.

**Keywords:** quality assessment; imbalanced dataset; classification accuracy; cropland area; map accuracy; image processing





## 1. Introduction

The Green Revolution caused widespread human interventions on the Earth's surface that had adverse impacts on local and global environmental ecosystems [1]. The authorities permit the transformation of native ecosystems for agricultural cultivation to feed the growing global population. A number of researchers and organizations now attribute changes to deforestation, industrialization, urban expansion [2,3], gains in agriculture and

losses in forests [4] and socio-economic development [5], primarily as a result of population growth in recent decades.

Land cover is known as the observed biophysical coverage of the ground surface, such as vegetation, bare soil, rock and built-up features on the Earth's surface [6], while land use is routinely derived from land cover and is related to local actions in the surrounding environment [7,8]. Land use/land cover (LULC) and related dynamics have been prioritized by the Global Land Project [9] as they essentially affect the biosphere, hydrosphere, atmosphere and lithosphere [10]. Reliable LULC dynamic maps can provide insight into human and ecosystem interactions [11,12] in an ever-changing world. Dynamic LULC changes are complex processes that show a degree of nonlinearity that is traceable to natural processes and anthropogenic activities [12]. Thus, such information is a preliminary key input for modeling crop type detection and crop rotation dynamics, which is very important for decision makers [13].

Data driven from any manual processing corrections is generally a time-consuming task and analyzing vast amounts of timely data is hard to manage. Scientists have proven that remotely sensed data is the best choice among other techniques to monitor LULC dynamic changes, considering its high spatiotemporal resolution, ready availability and wide coverage [14], as well as fast update speed [15]. Remotely sensed imagery needs to be spatially, spectrally and radiometrically processed to at least Level 2B in areas with low local relief. Ortho-rectification should also be accounted for in regions that are quite mountainous to obtain consistently high position accuracies.

Unsupervised classification and clustering were the initial common methods for LULC mapping at large-scales [16]. A wide variety of classification algorithms have been employed when researchers are faced with complex big data. The increasing availability of remotely sensed data in an era of big and open data provides new opportunities to reach an automated land cover classification [17]. Many of the current LULC maps have been produced mostly based on Machine Learning (ML), e.g., CGLS-LC 2019 (Copernicus Global Land Service) [18], IGBP DISCover (International Geosphere-Biosphere Program), UMD 1998 (The University of Maryland Department of Geography land cover classification), MODIS (Moderate Resolution Imaging Spectroradiometer) Land Cover, Global Land Cover (GLC 2000), GlobeCover 2009, and Global Land Cover by National Mapping Organizations (GLCMNO), and among them, Global Land Cover Datasets at a 30 m Resolution (Globeland30) [19] have the lowest spatial resolution (>100 m) at both continental and regional levels [20].

In this sense, a novel cloud computing platform named the Google Earth Engine (GEE) summarizes the data series that can be quickly recalled from Google servers with application programming interfaces (APIs) based on JavaScript and Python languages [21,22]. Unlike conventional LULC studies, GEE has a convenient coding system and high computing speed since it does not have the hardship of downloading raw satellite images from archives. From another point, access to a novel LULC service was developed by the advent of SAR imagery data, such as Sentinel-1. Therefore, optical and radar data fusion would considerably increase the accuracy of LULC mapping [23,24] and the monitoring of agricultural lands, starting with the launch of the ESA Earth observation mission in 2014.

Map producers need to assure potential users that their produced map is of high quality and accuracy assessment is a metric to this process [25]. Recently, the European Space Agency (ESA) WorldCover developed a land cover map V100 product at 300 m and 10 m resolutions based on Sentinel-1 and Sentinel-2 images for 2020 [26,27]. This new product on the GEE platform can significantly improve research efficiency by ensuring LULC mapping accuracy and reducing redundant studies that require LULC for modeling large-scale areas. In addition to the ESA product, FROM-GLC30 (Finer Resolution Observation and Monitoring of Global Land Cover) for 2017, Environmental Systems Research Institute (ESRI 2020) and GLC (Environmental Systems Research Institute 2020 Global Land Cover Map) maps are readily available. Finer Resolution Observation and Monitoring of Global Land Cover (FROM-GLC30) produced by Tsinghua University, China, is based on a

random forest algorithm [28], while ESRI, 2021 benefits from a deep learning model that was produced by the collaboration of ESRI and Microsoft [29].

The validation of the ESA product using grid-based sampling found an overall accuracy (OA) of 74.4 ± 0.1% and 80.7% for all 11 classes at the global and Asian scales [30]. In addition, applying a grid sampling scheme may result in losing data points for classes with a minimum area contribution and lead to an imbalanced training sample problem with an insufficient sample [31] as bioinformatics datasets are inherently imbalanced [32]. Hence, it should be noted that a balanced dataset can better demonstrate the performance of LULC classes. The findings showed that the ESA LULC CCI was the most accurate product on global and continental scales. However, investigation is needed to adjust ESA CCI LC 10 m spatial resolution via different sampling designs, i.e., balanced vs. imbalanced datasets based on the LACOVAL [33–35] standard on a more detailed scale in particular mountainous areas with high local relief variability employing an independent referenced dataset. LACOVAL is a validation tool for regional-scale land cover and land cover change that provides diverse accuracy metrics.

In view of the above, this research uses the ESA LULC CCI classification with support from the GEE platform and aims to evaluate the ESA LULC CCI product by employing four distinct spatial accuracy assessment strategies in the mountainous upper Karkheh River Basin (KRB), west Iran. Specifically, our objectives were to (1) explore the effects of a balanced dataset on the performance of LULC classes using the LACOVAL tool and (2) assess whether the accuracy of the product is equal to or better than the global and continental scales.

## 2. Materials and Methods

### 2.1. Study Area

The KRB has a semi-arid to arid climate and is one of the biggest rivers in Iran. The Honam sub-basin is located in the mainstream of the KRB bounded by 33°47′37″ N to 33°51′12″ N latitudes, 48°12′23″ E to 48°28′44″ E longitude with a broad altitude diversity ranged from 1519 m to 3575 m and an area of approximately 14,100 ha (Figure 1). The study area was selected on the basis that it represents nearly all major landforms. The heterogeneous landscape includes several types of land use, such as landscapes, which are mainly characterized by natural resources and human activities.

The region is characterized by arid, mild winter, warm to very warm summer based on agro-ecological zone (AEZ) of KRB according to De Pauw [36], meaning that it receives relatively low rainfall of 493.1 mm y$^{-1}$ (Alashtar synoptic meteorological station data during 2000–2021). Rainy season typically falls between October and April, whereas the dry season starts from April to September with negligible precipitation.

The region is generally an agriculture-based system in which livestock–food crop production farming is the leading financial activity of the local population. Besides, high quality agricultural lands have been converted and/or going to be converted to built-up due to growing land prices impressed by inflation in recent decades.

The ESA LULC CCI product was recalled from the GEE online platform [21]. Then, accuracy assessment was employed on independent ground truth to analyze the preliminary product. Since Gilmore Pontius Jr and Malizia [37] reported that map accuracy can be reduced or increased by land use category aggregation, the preliminary LULC were optimized based on ancillary data collected from several sources, taking into account temporal and thematic consistency.

An overview of the study procedure, which uses several techniques, is shown in Figure 2.

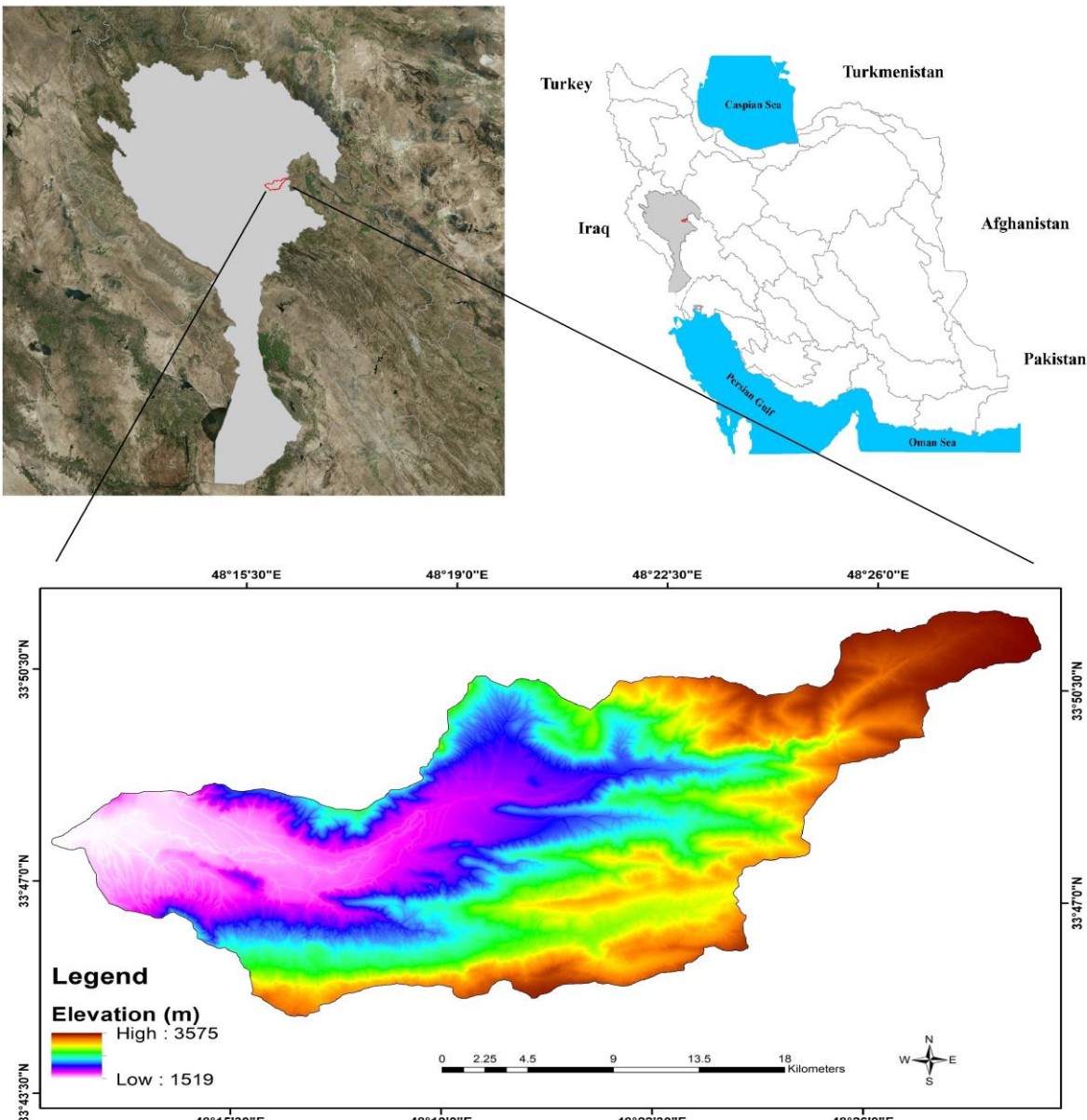

**Figure 1.** Location map of the study area over the Karkheh River hydrographic basin in south-west Iran.

### 2.2. Data Collection

European Space Agency Climate Change Initiative Land Cover (ESA CCI LC) is one of the most recent and detailed satellite products with 23 annual time series at the global level, which was released in October 2021. It is a free access global LULC product at 10 m resolution for a time span of 1992–2018, principally through Sentinel-1 and Sentinel-2 satellite images, containing 11 separated LULC classes [30]; six out of 11 LULC classes existing in the study region are summarized in Table 1. It should be pointed out that ESA has been providing another product with lower spatial resolution (300 m) from 1992 to 2015 that uses AVHRR/SPOT-VGT/MERIS sensors. This dataset divides yearly global maps into 22 classes, which have been defined based on the United Nations Food and Agriculture Organization's Land Cover Classification System (UN FAO-LCCS). The ESA WorldCover project envisions providing high-quality products to various users. Consequently, we expect that its developed novel services will be coming for accuracy assessment.

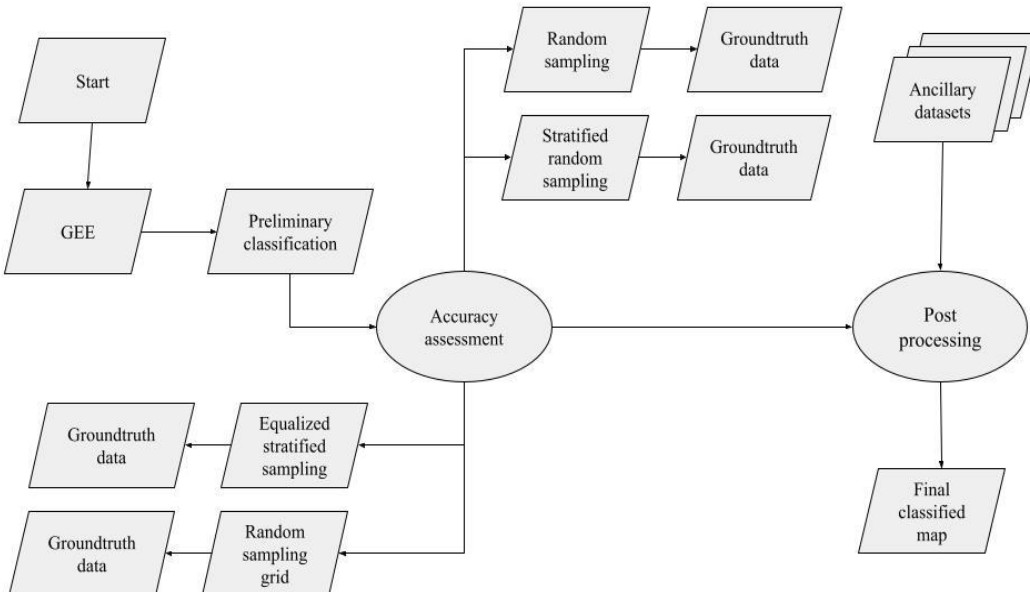

**Figure 2.** Workflow for accuracy assessment of ESA LULC CCI product.

*2.3. Geospatial Uncertainty Assessment*

The validation of satellite-driven products is needed during the remote sensing process since it gives the ability of perceptiveness to control the product quality, as well as users' confidence in using the satellite imagery products. In addition, the classification accuracy worsens with increasing heterogeneity and decreasing patch size, especially at coarser resolutions [38]. Therefore, the results of LULC classification need to be validated, as well as calibrated, where necessary [6]. It can be reached through several validation assessment methods, such as statistical, visual and spatial uncertainty assessment.

2.3.1. Statistical Assessment

Statistical assessment was carried out using the LACO-Wiki web-based framework for LULC validation, which was developed by the ESA-funded LandCover Validation (LACOVAL) standard [34,35] and the web architecture of GeoWiki [33]. LACO-Wiki has been designed user-friendly to facilitate the LULC validation procedure and to share multi-community users' LULC maps and referenced datasets [34]. It offers a map accuracy tool (tool.laco-wiki.net) that generates a square confusion matrix and similarly customized report on exploring thematic map accuracy metrics, choosing from a set of different quality indicators, including overall accuracy (OA), user's accuracy (UA), producer's accuracy (PA), kappa (KA), average mutual information (AMI), portmanteau (PMA), partial portmanteau (PMAP), allocation disagreement (All.Dis), quantity disagreement (Quan.Dis), shift, exchange, average mutual information (AMI) and adjusted average mutual information (Adj.AMI), except average user's accuracy (AUA) average producer's accuracy (APA), and average partial portmanteau (Ave.PMAP). The details are summarized in Table 2.

**Table 1.** Overview of the ESA LULC CCI product classes of our study based on the integration of FROM-GLC30 [28] and ESRI 2020 [29] Land Cover legends.

| Class | Code | Description |
|---|---|---|
| Trees | 10 | Any significant clustering of tall (~15-m or higher) dense vegetation, typically with a closed or dense canopy; examples: wooded vegetation, clusters of dense tall vegetation within savannas, plantations, swamp or mangroves (dense/tall vegetation with ephemeral water or canopy too thick to detect water underneath). |
| Shrubland | 20 | Shrubland is mix of small clusters of plants or single plants dispersed on a landscape that shows exposed soil or rock; scrub-filled clearings within dense forests that are clearly not taller than trees; examples: moderate to sparse cover of bushes, shrubs, and tufts of grass, savannas with very sparse grasses, trees or other plants. Its cover has a texture finer than tree canopies but coarser than grasslands. With a height between 0.3 and 5 m and cover percentage >15%. |
| Grassland | 30 | Open areas covered in homogenous grasses with little to no taller vegetation; wild cereals and grasses with no obvious human plotting (i.e., not a plotted field); examples: natural meadows and fields with sparse to no tree cover, open savanna with few to no trees, parks/golf courses/lawns, pastures; grasslands for grazing, as well as natural grasslands; herbaceous cover percentage classification > 15%. |
| Cropland | 40 | Human planted/plotted cereals, grasses, and crops not at tree height; examples: corn, wheat, soy, fallow plots of structured land. Land that has clear traits of intensive human activity. It varies from bare field, seeding, crop growing, to harvesting. Fruit trees are classified into forests. Pasture could be transitional from croplands to natural grasslands. Lands for rice cultivation, arable and tillage lands, greenhouse farming. |
| Built-up | 50 | Human-made structures; major road and rail networks; large homogenous impervious surfaces, including parking structures, office buildings and residential housing; examples: houses, dense villages/towns/cities, paved roads and asphalts. |
| Barren land | 60 | Lands with very sparse to no vegetation or not covered by vegetation or vegetation is hardly observable for the entire year, but dominated by exposed soil, sand, gravel and rock backgrounds. Dry salt flats/pans occurring on the flat floored bottoms of interior desert basins; dried lake beds, mines; sandy areas composed primarily of dunes; gravel land and bare rocks; other types of land not covered by vegetation or with no to little vegetation, such as lake/river bottoms, in the dry season. |

**Table 2.** Statistics used in this study for validation of ESA LULC CCI map.

| Statistics | Acronym | Definition | Reference |
|---|---|---|---|
| Overall accuracy | OA | Calculated as the total number of correctly classified pixels (diagonal elements) divided by the total number of test pixels. | [39] |
| User's accuracy | UA | The fraction of correctly classified pixels with regard to all pixels selected as a given class. | [40] |
| Producer's accuracy | PA | The fraction of correctly classified pixels with regard to all pixels of a given ground truth class. | [40] |
| Kappa | KA | A statistic that measures inter-rater agreement for qualitative (categorical) items. It generally takes into account the agreement occurring by chance. | [41,42] |
| Average mutual information | AMI | Measuring the dependence between two variables. AMI provides a means of assessing the similarity of maps with different themes, i.e., the amount of information that one map predicts of the other. | [43,44] |
| Portmanteau | PMA | Describes the overall accuracy when the data are collapsed to two classes, the land cover type of interest, and all other land cover types combined into a single class. | [45] |
| Partial portmanteau | PMAP | Known as "figure of merit" and is robust to the source of bias. | [45] |
| Allocation disagreement | All.Dis | The amount of difference between the reference map and a comparison map that is due to the less than optimal match in the spatial allocation of the categories, given the proportions of the categories in the reference and comparison maps. | [46] |
| Quantity disagreement | Quan.Dis | The amount of difference between the reference map and a comparison map that is due to the less than perfect match in the proportions of the categories. | [46] |
| Shift | Shift | Exists for a pair of pixels when one pixel is classified as category A in the first map and as category B in the second map, while simultaneously the paired pixel is classified as category B in the first map and as category A in the second map. | [47] |
| Exchange | Exchange | Exists for more than two categories of pixels, which are the allocation differences that are not exchanged. | [47] |
| Average mutual information | AMI | The amount of information shared between a set of classified and reference points. | [43] |
| Adjusted average mutual information | Adj.AMI | Normalized AMI to the theoretical maximum amount of information possible given the distribution of categories in a map. | [25] |
| Average user's accuracy | AUA | An average of the accuracy of individual categories of the user's accuracy. | [48] |
| Average producer's accuracy | APA | An average of the accuracy of individual categories of the producer's accuracy. | [48] |
| Average partial portmanteau | Ave.PMAP | An average of individual categories of the partial portmanteau. | - |

### 2.3.2. Spatial Post-Classification Accuracy Assessment

Widely known statistical accuracy assessment values, such as kappa, or overall accuracy give a general idea about the accuracy of thematic maps [30]. It does not provide any spatial interpretation against map quality. Spatial accuracy parameters inform users regarding magnitudes of the source estimates [49] and degree of uncertainty in LULC mapping across space. These types of spatially accurate estimates contribute to a better choice of the most probable LULC maps for a region of interest. In addition, LULC product accuracy should be estimated over a significant set of locations (typically > 30) with reference to in situ data [50].

The assembly of appropriate sampling design for the LULC referenced dataset was one of the main challenges for the assessment of the product reliability as the assessment procedure highly relies on quantity, quality and availability of reference points. Given the imbalanced sampling density between and within classes, absence, over- and under-sampling can considerably influence the outcomes, four separate sampling schemes (Figure 3) for a total of 180 reference data points (Table 3) were investigated on map quality as follows:

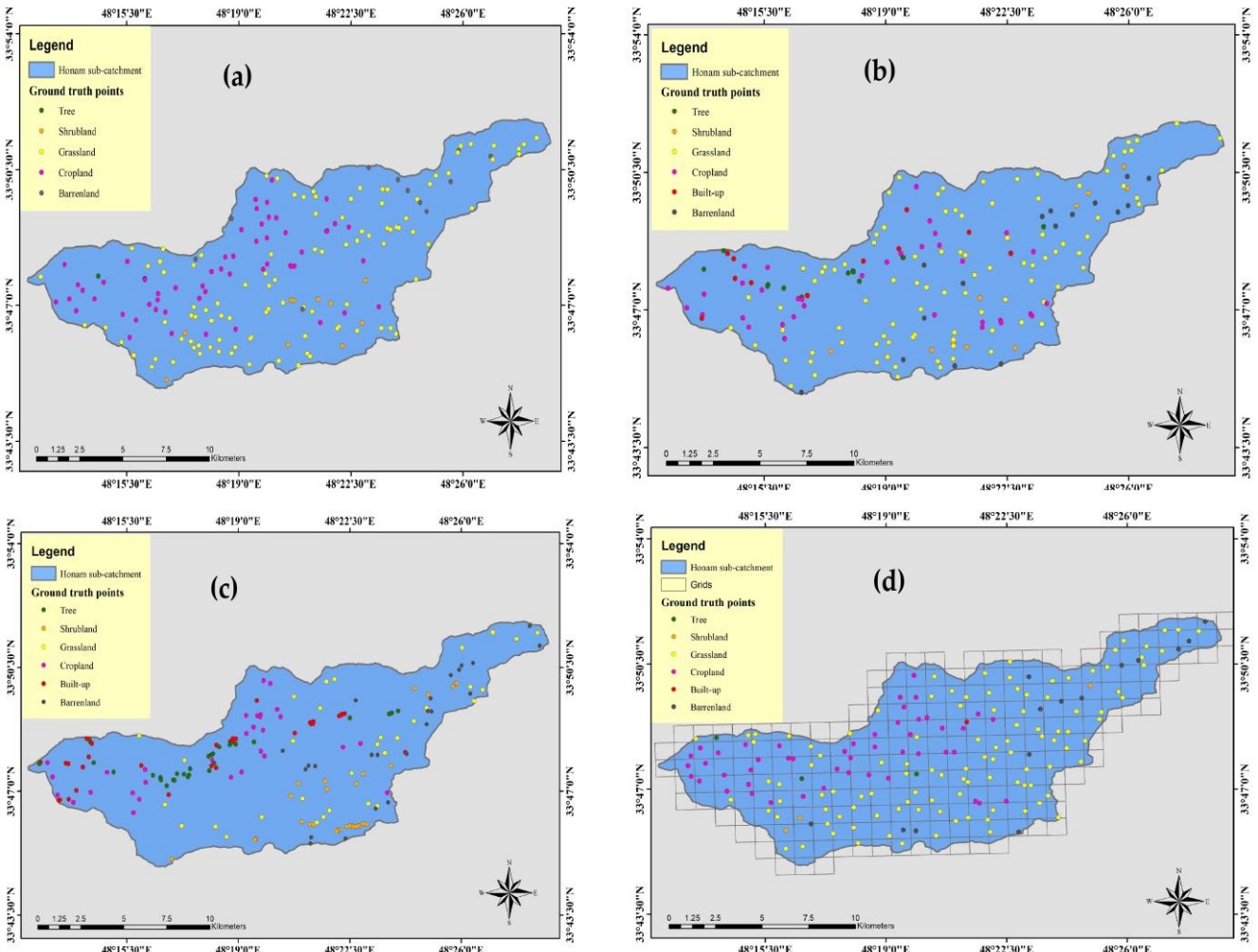

**Figure 3.** Spatial distribution of validation samples over the Honam sub-basin: (**a**) random (R), (**b**) stratified random (SR), (**c**) equalized stratified random (ESR) and (**d**) random sampling grid (RSG) sampling schemes.

Sampling scheme 1: Random sampling (RS)
A total of 180 points are randomly distributed over the study region, neglecting LULC classes. Duplication is not possible.
Sampling scheme 2: Stratified random sampling (SRS)
A total of 180 points are randomly distributed over the six specified classes (known as strata).
Sampling scheme 3: Equalized stratified random sampling (ESRS)
A total of 180 random points are evenly distributed in each LULC class (30 distinct random referenced dataset for six classes).
Sampling scheme 4: Random sampling grid (RSG)
A total of 180 random points are distributed based on 873 × 873 m grids.

**Table 3.** Distribution of the number and portion of reference points collected from different sampling schemes.

| Scheme | | LULC Class | | | | | | |
|---|---|---|---|---|---|---|---|---|
| | | **Tree** | **Shrubland** | **Grassland** | **Cropland** | **Built-Up** | **Barren Land** | **Total** |
| 1 | No. of samples | 0 | 1 | 120 | 52 | 0 | 7 | 180 |
| | Portion (%) | 0 | 0.5 | 66.7 | 29 | 0 | 3.8 | 100 |
| 2 | No. of samples | 13 | 10 | 95 | 35 | 11 | 16 | 180 |
| | Portion (%) | 7.2 | 5.6 | 52.8 | 19.5 | 6 | 8.9 | 100 |
| 3 | No. of samples | 30 | 30 | 30 | 30 | 31 | 29 | 180 |
| | Portion (%) | 16.7 | 16.7 | 16.7 | 16.7 | 17.1 | 16.1 | 100 |
| 4 | No. of samples | 0 | 0 | 129 | 46 | 1 | 4 | 180 |
| | Portion (%) | 0 | 0 | 71.7 | 25.6 | 0.6 | 2.1 | 100 |

2.3.3. Visual and Survey Assessment

The ESA LULC CCI product was compared visually using a variety of reference layers, including VHR satellite and aerial imagery from Google and Bing, as well as OpenStreetMap (latest version of LACO-Wiki of March 2020). For this purpose, GeoWiki provides an online platform for users to upload the LULC map, choose sampling design of random point, random pixel, random polygon, polygon at random point, stratified random point, stratified random pixel and stratified random polygon (LACO-Wiki quick start guide). The LACO-Wiki validation procedure contains a simple four-step process: (1) upload dataset, (2) generate validation samples, (3) validate the map and, and (4) create a report.

Sampling schemes 3 and 4 were added manually in ArcGIS 8 since LACO-Wiki does not represent equalized and grid-based sampling schemes in its supported sampling methods. In addition, several field revisits were conducted during 2020–2021 to identify all suspicious points of each class via in situ reference data. In some areas, additional existing historical Google Earth images and other suitable referenced datasets were also used to improve the quality of the final map. In the case of any change in LULC classes, the information was updated accordingly.

2.3.4. Post-Processing and Map Validation

Following the accuracy assessment of the LULC product, post-processing was carried out to check and correct the LULC map for further studies. For this purpose, pixels in the LULC classification map were converted to a vector format to relabel the values. The map was then validated by confirming the class or by changing those classified incorrectly in LACO-Wiki (Merged matrix tool). Finally, existing peer-reviewed ancillary datasets were integrated with raw ESA LULC CCI product to improve the accuracy of the final classified map; this is specifically suggested for identifying missed categories and/or classes with less area portion, such as roads and rural buildings. These rules-based corrections were applied based on the fact that certain LULC classes match with that of the historical ones.

**3. Results**

*3.1. Preliminary LULC*

The ESA LULC CCI thematic map was exported from GEE. The output 10 m resolution map depicts 6 out of 11 LULC CCI classes as trees, shrublands, grasslands, croplands, built-up and barren lands. This was in agreement with field survey results, to a significant extent, except for river as water body class. An overview of the output map is given in Figure 4, whereas individual calculated components of LULC are registered in Table 4. Official data refer to the year 2020, in which 94% of the sub-basin area, already disregarding other land uses, belongs to agricultural activities and livestock use.

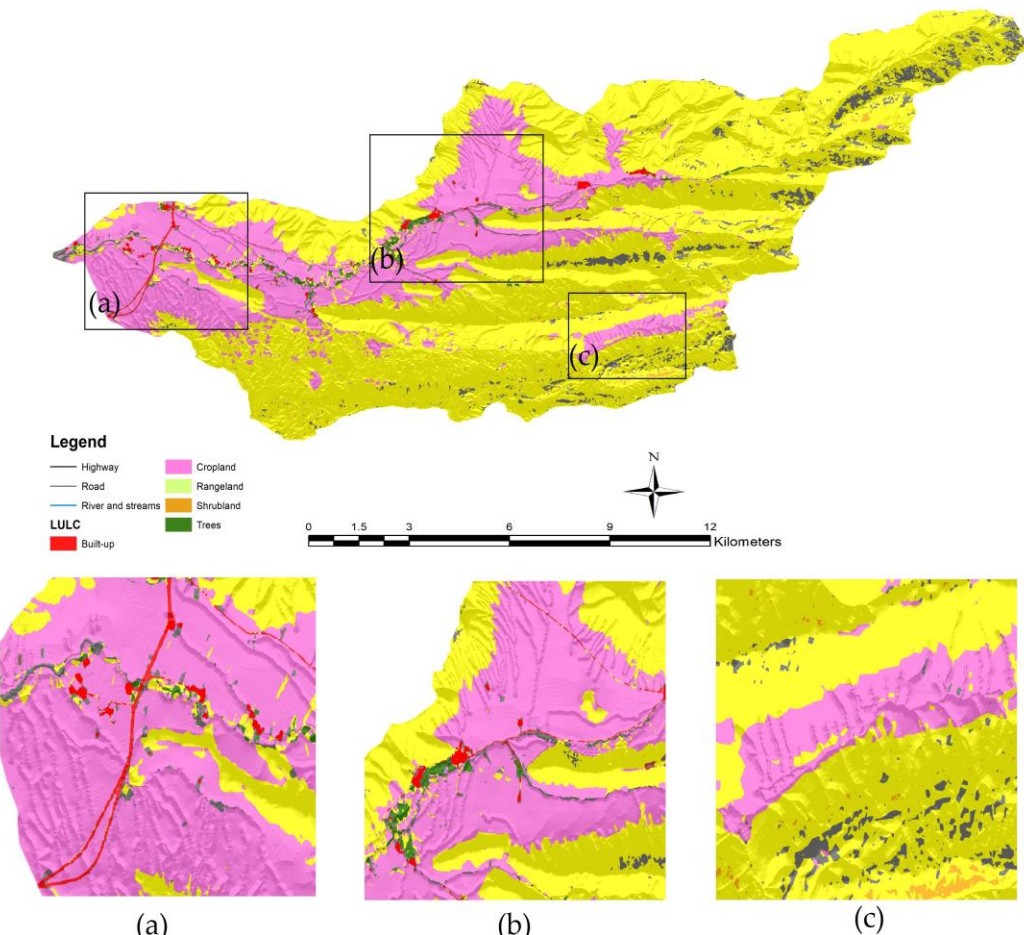

**Figure 4.** Land use–land cover map of the Honam sub-basin extracted from ESA LULC CCI 2020 product that coded on the Google Earth Engine platform. Zoomed overview from: (**a**) western part, (**b**) central and (**c**) eastern part of the region of interest. Legend colors are compatible with those proposed by the ESA LULC CCI product.

**Table 4.** Land use and land cover distribution of the Honam sub-basin derived from ESA LULC CCI.

| LULC Class | Extent (ha) | Portion of Area (%) |
|---|---|---|
| Tree | 72 | 0.5 |
| Shrubland | 37 | 0.3 |
| Grassland | 9775 | 69.8 |
| Cropland | 3411 | 24.2 |
| Built-up | 91 | 0.6 |
| Barren land | 654 | 4.6 |

Figure 4 shows that croplands are present in central/south-west regions of the sub-basin, where the elevation ranges from 1519 m to 2875 m. This exactly matches the purple regions of the digital elevation model (Figure 1). The tree class is dispersed around water resources, such as the Honam perennial river, which is regularly located at the lowest parts of the sub-basin (Figure 4b). Most of the shrubland class is found in the southern mountains of the study area (Figure 4c) at the lower elevation range limits, compared to croplands. Urbanization mainly developed along the river and roads (Figure 4a).

The results from the classified map indicated that 69.8% of the sub-basin area was covered by grassland, followed by cropland (24.2%), barren land (4.6%), built-up (0.6%), tree (0.5%), and shrubland (0.3%).

*3.2. Spatial Accuracy Assessment*

3.2.1. Random Sampling (RS)

Accuracy estimates of the random sampling scheme can be found in Table 5. On a regional scale, the overall map accuracy was 87%. In terms of class-specific accuracies, grassland and cropland had high accuracies according to UA and PA. Shrubland and barren land classes had moderate accuracies, while tree and built-up had zero accuracy due to the lack of reference data within the classes (Figure 3a). In general, there was an overestimation of the grassland compared to the random validation dataset.

**Table 5.** Confusion matrix and quality indicators of the random sampling scheme for the ESA LULC CCI product.

| | Observation | | | | | | |
|---|---|---|---|---|---|---|---|
| **Predicted** | **Trees** | **Shrubland** | **Grassland** | **Cropland** | **Built-Up** | **Barren Land** | **UA** |
| Trees | - | - | - | - | - | - | 0% |
| Shrubland | - | 1 | - | - | - | - | 100% |
| Grassland | - | 9 | 101 | 4 | - | 6 | 84.2% $\pm$ 0.065 |
| Cropland | 2 | - | - | 50 | - | - | 96.2% $\pm$ 0.052 |
| Built-up | - | - | - | - | - | - | 0% |
| Barren land | - | - | - | 2 | - | 5 | 71.4% $\pm$ 0.36 |
| PA | 0% | 56.9% | 100% | 89.21% | 0% | 60.87% | 87% |
| PMA | 98.79% | 95.74% | 91.02% | 95.13% | 100% | 95.4% | |
| PMAP | 0% | 10% | 84.17% | 86.21% | 0% | 38.46% | |

Note: correct classifications are on the matrix diagonal; misclassifications are off-diagonal [51], UA: user's accuracy, PA: producer's accuracy, MAP: portmanteau, PMAP: partial portmanteau.

The calculated portmanteau (PMA) values were other than zero for classes in which no sample exists. Surprisingly, the values for the remaining classes were almost the same for various UA and PA, ranging from 56.9% to 96.2%. The partial portmanteau (PMAP) metric was more robust than the PMA to the source of bias that OA has nothing to do with the class of interest [45] because PMAP eliminates true negatives from the calculation unlike PMA. This has been meaningful in our study with 0% PMAP of no-sampled classes. The results of the PMAP showed that the values were close to the UA estimates. In addition, cropland and grassland obtained a high accuracy of 86.21% and 84.17%, respectively. However, the accuracy was strongly adjusted for shrubland (from 56.9% to 10%) and barren land (from 60.87% to 38.46%).

3.2.2. Stratified Random Sampling (SRS)

The square confusion matrices and class-specific accuracies for the SRS sampling scheme are listed in Table 6. Similar to RS, OA reached 87%. It happened on occasion that often-neglected classes, i.e., tree, built-up and shrubland, had around 10 samples per class since strata were formed based on classes' shared attributes in SRS.

The accuracy of the most prevalent class of the study region, i.e., grassland (see Table 4), increased by 9.48%, while the accuracy of cropland decreased by 10.49% in terms of UA. However, the PA metric tended to be improved for these classes.

According to PMAP, built-up and grassland could achieve reasonable accuracy, while tree, cropland and shrubland classes had moderate accuracy; barren land again had low accuracy.

3.2.3. Equalized Stratified Random Sampling (ESRS)

The RS and SRS sampling schemes suggest unequal sample inclusion probabilities with fixed sample size [52,53]. A key factor for the accurate assessment of LULC classification is a sufficient amount of high-quality reference data, spanning all detectable classes with relatively balanced observations and distribution [31,32]. Since the ESA LULC CCI detected

six classes for the region of interest, ESRS allows for the spatial distribution of the reference data with a roughly equal set of 30 in situ locations for each class according to the suggestion of Bayat [50].

**Table 6.** Confusion matrix and quality indicators of the stratified random sampling scheme.

| | Observation | | | | | | |
|---|---|---|---|---|---|---|---|
| **Predicted** | **Trees** | **Shrubland** | **Grassland** | **Cropland** | **Built-Up** | **Barren Land** | **UA** |
| Trees | 9 | - | - | 3 | - | 1 | 69.23% ± 0.26 |
| Shrubland | - | 8 | 2 | - | - | - | 80% ± 0.26 |
| Grassland | - | 2 | 89 | 3 | - | 1 | 93.68% ± 0.05 |
| Cropland | - | - | 4 | 30 | - | 1 | 85.71% ± 0.11 |
| Built-up | 1 | - | - | - | 10 | - | 90.91% ± 0.17 |
| Barren land | - | - | 9 | - | - | 7 | 43.75% ± 0.25 |
| PA | 88.39% ± 0.2 | 78.51% ± 0.24 | 89.25% ± 0.04 | 84.66% ± 0.1 | 100% | 60.2% ± 0.3 | 87% |
| PMA | 97.79% | 97.67% | 89.83% | 94.04% | 99.49% | 95.27% | |
| PMAP | 64.29% | 66.67% | 80.91% | 73.17% | 90.91% | 36.84% | |

Note: correct classifications are on the matrix diagonal; misclassifications are off-diagonal [51], UA: user's accuracy, PA: producer's accuracy, MAP: portmanteau, PMAP: partial portmanteau.

Half of the LULC classes received exactly 30 reference data points; grassland, tree and built-up classes were over-sampled, while barren land was under-sampled (Table 7). This can be attributed to the proximity of tree and built-up samples that had a very low area contribution as compared to remaining 98.6%, which concerns the high probability of neighboring samples replacing within the neighboring classes (Figure 3c). The shrubland did not follow over- and under-sampling due to its geographical distance from other classes. The ESRS sampling scheme resulted in an accuracy as high as 91.6%.

**Table 7.** Confusion matrix and quality indicators of the equalized stratified random sampling scheme.

| | Observation | | | | | | |
|---|---|---|---|---|---|---|---|
| **Predicted** | **Trees** | **Shrubland** | **Grassland** | **Cropland** | **Built-Up** | **Barren Land** | **UA** |
| Trees | 30 | - | - | - | - | - | 100% |
| Shrubland | - | 27 | 3 | - | - | - | 90% ± 0.1 |
| Grassland | - | 2 | 26 | 1 | - | 1 | 86.67% ± 0.12 |
| Cropland | - | - | 1 | 29 | - | - | 96.67% ± 0.065 |
| Built-up | 1 | - | - | - | 30 | - | 96.77% ± 0.063 |
| Barren land | - | 1 | 2 | - | 3 | 23 | 79.31% ± 0.15 |
| PA | 96.88% ± 0.06 | 89.6% ± 0.1 | 82.21% ± 0.11 | 96.45% ± 0.067 | 90.91% ± 0.09 | 95.57% ± 0.08 | 91.6% |
| PMA | 99.45% | 96.65% | 94.39% | 98.87% | 97.81% | 96.14% | |
| PMAP | 96.77% | 81.82% | 72.22% | 93.55% | 88.24% | 76.67% | |

Note: correct classifications are on the matrix diagonal; misclassifications are off-diagonal [51], UA: user's accuracy, PA: producer's accuracy, MAP: portmanteau, PMAP: partial portmanteau.

By balancing the dataset specifically for barren land, it was possible to reach a higher accuracy of 98%, the highest accuracy of our study. We point out that cropland accuracy became very ideal according to both high PA and UA values. Tree classification accuracy alone reached approximately 97%, whereas the accuracy of this sampling scheme was slightly higher than that of RS and SRS.

As reported by PMAP, accuracy was sharply increased, with 17.5% and 39.8% improvement for shrubland and barren land classes, respectively, in the ESRS sampling scheme.

This could be related to either a higher sample size compared to the previous or PMAP true negative elimination.

### 3.2.4. Random Sampling Grid (RSG)

The gridded 873 × 873 m sampling scheme covers similar 180 reference points and showed a reasonable spatial distribution pattern (Figure 3d) over the study area. The RSG yields a considerably higher accuracy than RS, SRS and ESRS, only referring to UA and PA. However, it reaches 98% accuracy; the classes occupying only a small proportion of a landscape outperform zero accuracy for the same reason as stated for RS (Table 8). The PMAP values also demonstrated this.

**Table 8.** Confusion matrix and quality indicators of the random sampling grid.

| | Observation | | | | | | |
|---|---|---|---|---|---|---|---|
| **Predicted** | **Trees** | **Shrubland** | **Grassland** | **Cropland** | **Built-Up** | **Barren Land** | **UA** |
| Trees | - | - | - | - | - | - | 0% |
| Shrubland | - | - | - | - | - | - | 0% |
| Grassland | - | - | 127 | 1 | - | 1 | 98.45% ± 0.02 |
| Cropland | - | - | - | 46 | - | - | 100% |
| Built-up | - | - | - | - | 1 | - | 100% |
| Barren land | - | - | 1 | - | - | 3 | 75% ± 0.49 |
| PA | 0% | 0% | 99.21% | 97.93% | 100% | 75.15% | 98% |
| PMA | 100% | 100% | 98.34% | 99.45% | 100% | 98.89% | |
| PMAP | 0% | 0% | 97.69% | 97.87% | 100% | 60% | |

Note: correct classifications are on the matrix diagonal; misclassifications are off-diagonal [51], UA: user's accuracy, PA: producer's accuracy, MAP: portmanteau, PMAP: partial portmanteau.

Better characterization of the ESA LULC CCI product at a regional scale compared to global validation [30] might be related to its inherent level of high spatial resolution, which will surely be neglected at a global scale. High overall accuracy seems to be misleading when additional points not in that category are correctly classified. We expect that more reference data were placed for tree, shrubland and built-up classes in RSG, especially against RS and ESRS designs, because sample inclusion probability is bounded to squared 873 m in this approach.

We found that a high correlation existed between the number of reference data and the share of the area. For example, 127 out of 180 grassland reference data were equal to 69.8% of area (for more information on this data, see Table 4). This approved that RSG allocated samples into classes according to the corresponding area's share. The accuracy of barren land was comparable to those previously calculated sampling schemes, except for ESRS, in terms of UA and PA metrics. However, PMAP represented 16.67% lower accuracy in RSG than ESRS. In the case of adding the PMAP metric to the quality indicators by Tsendbazar [30], the non-random geographical distribution was more apparent.

## 4. Discussion

For individual LULC classes, PA and UA provided partly related perspectives on category-level accuracy. Interestingly, in an accuracy metric intensive environment, the proposed equalized design, i.e., ESRS, seemed to be much more efficient in the accuracy assessment of individual LULC classes (see red lines in Figure 5), as our intuitive sense was raised. This is in consistent with Mallah [32] who reported that the use of balanced data in-creases the OA and Kappa metrics of soil texture classes.

It had higher accuracy than others, except for built-up. Although the built-up class had lower accuracy compared to other LULC classes, the prediction results of Fitton [54] showed an accuracy of 72% for the road class

Radar pattern of accuracy metrics noted that ESRS performed better for cropland (Figure 5d) and grassland (Figure 5c) compared to other sampling schemes of this study; RSG was at the lowest level of precision, mainly due to their much higher sample size. For instance, SRS and RSG embraced 70.5% and 57.7% of the total reference data in grassland class, while there were no differences between the sampling schemes for built-up (Figure 5e). Inspecting the individual sampling design, accuracy metrics had a tendency toward worsening when the reference data were more randomly distributed among classes, notably for trees (Figure 5a) and built-up (Figure 5e) land uses, due to their small width relative to the tile size [54].

Map producers are regularly interested in RSG sampling design because they desire to create multi-purpose thematic maps. As demonstrated below, highlighting a simple accuracy assessment would result in misinterpretation or misuse, but a comprehensive approach will give deep insight to the users for the benefits of detailed LULC information. Specifically, map users who focus on one or a few LULC classes are seriously recommended to apply class-level rather than map accuracy metrics.

During our visual comparisons, we found that, despite the relatively rational differentiation between the nearest LULC classes (see Figure 6b,d), there was some anomalies in the product. For example, visible rocks were mostly misclassified as barren land (see the yellow lines in Figure 6c). This land-use class is mainly considered rocks and outcrops in agricultural and natural resource terminologies of arid and semi-arid mountainous regions [55]. No vegetation can grow on rocky outcrops because it is an exposure of underlying bedrock or ancient consolidated deposits on the Earth's surface.

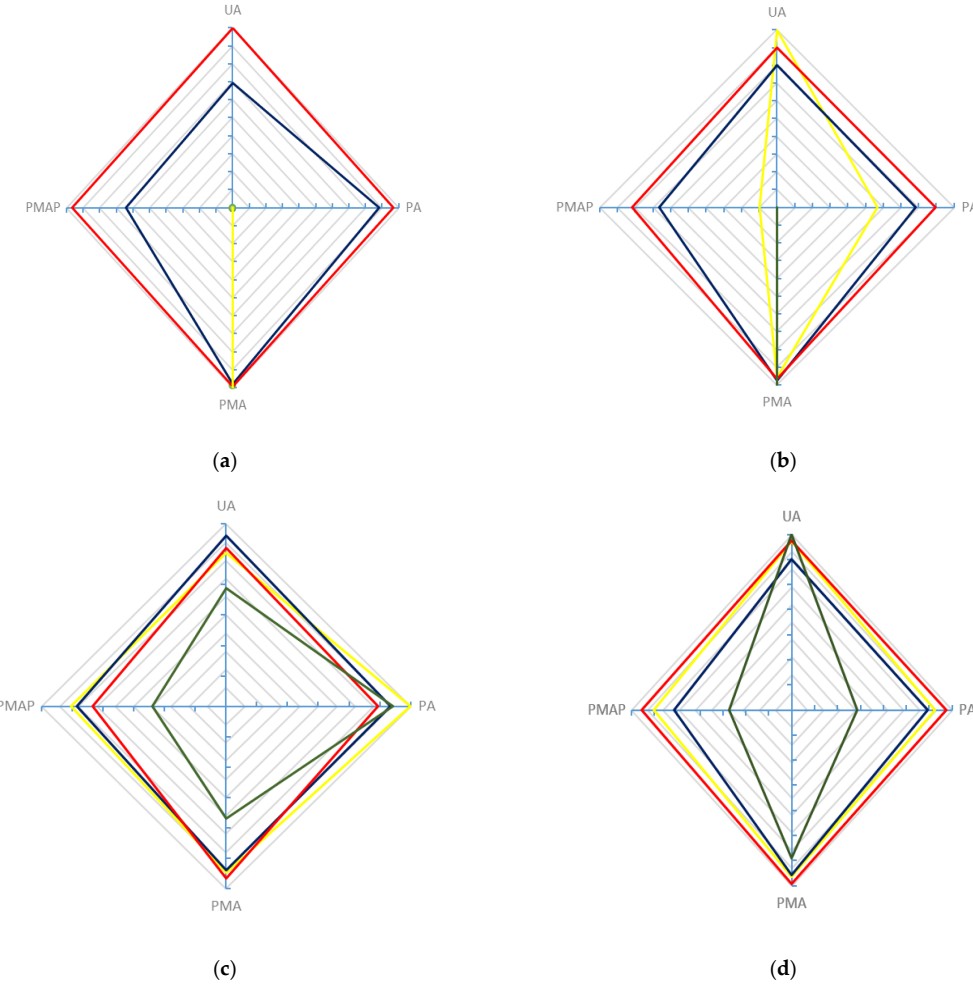

**Figure 5.** *Cont.*

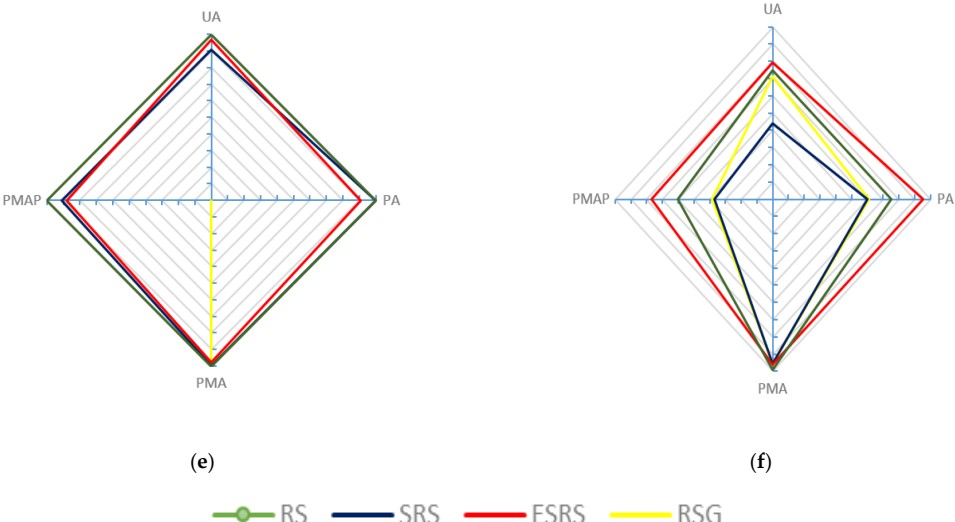

**Figure 5.** Radar charts regarding accuracy metrics for each LULC class with respect to four sampling schemes: (**a**) tree, (**b**) shrubland, (**c**) grassland, (**d**) cropland, (**e**) built-up and (**f**) barren land. RS: random sampling; SRS: stratified random sampling; ESRS: equalized stratified random sampling; RSG: random sampling grid.

Shrublands are not believed to be a major category of LULC in Iran, but their application is quite dissimilar to grassland and forest. The shrubs of the study area are typically sparse coarse-textured hawthorn covers that are normally smaller than mature trees (see description of Table 1). We realized that shrubs have geographically distributed patterns. Distance between two shrubs ranged from 10 m to 12 m (Figure 6a). However, the formed clusters were mostly farther apart.

Similar to rocky outcrop anomalies, grassland and barren land classes have different definitions in drylands, which are not compatible with the context of the local community. Due to frequent drought, grass is established only in the three months of spring, and lands are usually bare in other seasons. Thus, experts replaced those with rangeland LULC classes. Rangelands are characterized as vast natural landscapes where livestock and wildlife graze on natural vegetation or climax with complex functional interactions [56]. It is differentiated from pasture that cattle are kept basically in managed agricultural lands for feeding.

Each quality indicator has particular advantages and drawbacks. Likewise, the type of class-specific metrics is complicated, particularly when one or more samples over class under consideration are rare. Generally, the accuracy increases when more data are added to a specific class. In our study, RSG and ESRS had reasonable accuracy based on OA and KA metrics (Table 9). Fitton [54] could not reach accuracy greater than 90% for each class using the developed convolutional neural network (CNN) model, while Jin and Mountrakis [23] provided the maximum OA of 83% for different land cover types by combining spectral, scattering and vertical structural information.

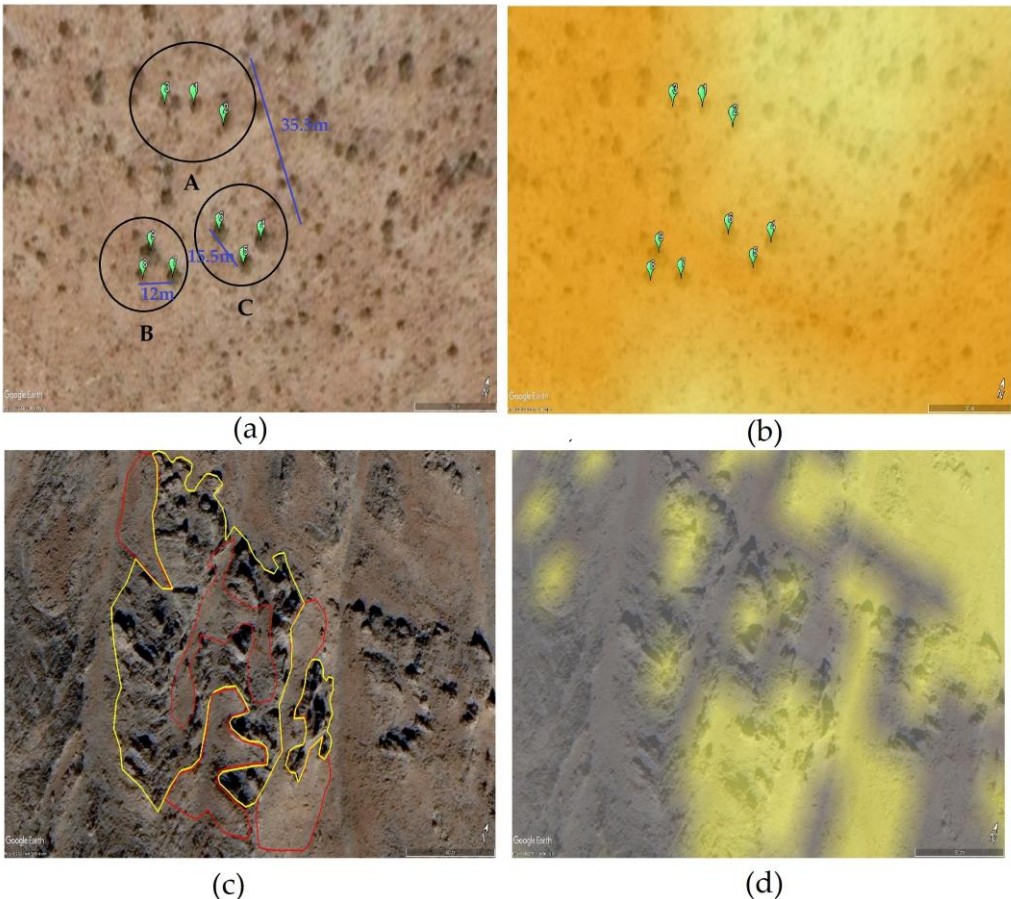

**Figure 6.** Comparison of real true-color composition satellite image with ESA LULC classes: (**a**) a screenshot of a Google image taken from areas with typic shrubland LULC class as a sample for interpreta-tion; green points are mostly Hawthorn shrubs; A, B and C circles are the groups of shrubland; blue lines are distances between shrubs and clusters, (**b**) shrubland classification map of ESA LULC, (**c**) a screenshot of Google image taken from areas with typic barren land LULC class as a sample for interpretation; red polygons are rangeland in spring and bare land at the remaining time of the year; yellow polygons are rock and outcrop, and (**d**) barren land classification map of the ESA LULC.

**Table 9.** Error metrics comparison of the sampling scheme.

| Sampling Scheme | Indicator | | | | | | | | | | |
|---|---|---|---|---|---|---|---|---|---|---|---|
| | OA | KA | AUA | APA | Ave.PMAP | All.Dis | Shift | Exchange | Quan.Dis | AMI | Adj.AMI |
| RS | 0.87 ± 0.046 | 0.76 ± 0.085 | 58.6% | 51.2% | 36.47 | 0.029 | 0.028 | 0 | 0.08 | 1.1 ± 0.157 | 1.08 |
| SRS | 0.87 ± 0.045 | 0.76 ± 0.08 | 77.2% | 83.5% | 68.8 | 0.098 | 0.026 | 0.07 | 0.03 | 1.16 ± 0.22 | 1.15 |
| ESRS | 0.91 ± 0.039 | 0.91 ± 0.04 | 91.6% | 91.9% | 91.9 | 0.056 | 0.01 | 0.04 | 0.026 | 2.13 ± 0.16 | 2.13 |
| RSG | 0.98 ± 0.018 | 0.96 ± 0.04 | 62.2% | 62% | 59.3 | 0.011 | $4.3 \times 10^{-5}$ | 0.01 | 0.005 | 0.91 ± 0.15 | 0.91 |

RS: random sampling, SRS: stratified random sampling, ESR: equalized stratified random sampling, RSG: random sampling grid, OA: overall accuracy, AUA: average user's accuracy, APA: average producer's accuracy, Ave.PMAP: average partial portmanteau, All. Dis: allocation disagreement, Quan.Dis: quantity disagreement, AMI: average mutual information, Adj.AMI: adjusted average mutual information, KA: kappa.

The ESRS worked more efficiently due to the fact that the lower the OA and KA differences, resulted in a higher precision. Researchers attempt to solve this problem by introducing the kappa metric for the chance agreement of observed versus referenced datasets [42]. Indeed, OA is a map-level metric, while portmanteau and kappa under certain circumstances are the category-level statistics by collapsing the confusion matrix into a binary matrix for each class [25].

We criticized that the OA could not appropriately describe the accuracy of the LULC map, in which OA is the proportion of the sum of the confusion matrix diagonally divided by the sum of the total confusion matrix. Actually, overall accuracy is not a good quality indicator for the reason that rare-sampled classes i.e., built-up, tree and shrubland, are not entering into calculation. OA is a suitable metric to make a fairly simple comparison between competing maps whenever competing maps share a common class's type and number of categories [53], such as land cover change detection maps.

The average weighted UA is equal to OA when the values are weighted by the classified (user's) frequencies [25]. Here, we found that the equally weighted average UA and PA, known as average user's (AUA) and average producer's (APA) balanced accuracy rates [44,57–59] better explain biases for a simple reason that involves zero accuracy classes in the calculation.

Although KA is not encouraged for many reasons, such as not adding any fundamental information to basic accuracy assessment, Pontius and Millones [46] proposed novel metrics named allocation disagreement (All.Dis) and quantity disagreement (Quan.Dis), as alternatives to KA, which is a disagreement measure between reference points and corresponding classified one. Accordingly, RSG showed better results since All.Dis and Quan.Dis of RSG obtained the least values (Table 9).

Overall, the accuracy calculation is highly dependent upon the chosen validation points. Specifically, if the samples are not representative of the whole landscape, the calculated value will be biased [60]. However, inspecting the error metric values, calculated KA, MPA, AUA and Ave.PMAP were found to be approximately the same as the overall accuracy of 0.91; this implies appropriate sampling design selection.

*Validated Map*

As stated by Pontius and Millones [46], the OA of the newly produced map must be at least equal or better than the original once, our combination of grassland and barren land categories also proven that all accuracy metrics improved significantly. For a given combination of grassland-barren land, there is no way for correcting reference points to become incorrect [25].

Overall, accuracy increased from 91% $\pm$ 0.039 in the original ESA LULC CCI map, up to 93.5% $\pm$ 0.04 in the validated map (Table 10). Our assessment revealed that the OA of ESA LULC CCI tended to be improved once downscaled from the global to continental level (Asia) by 6.7% $\pm$ 0.1 [30], from the continental to regional level by 12.8 $\pm$ 0.06 and from the global to regional level by 19.1% $\pm$ 0.06. However, Gong [28] published a lower OA of 72.76% for 10 common LULC classes, and ESRI 2021 product reported a higher OA of 86% for eight LULC classes, excluding snow/ice class and persistent cloud cover areas. Hermosilla [17] had proven that regionalization of the model ensured locally relevant descriptors and resulted in improved classification outcomes from OA of 70.3 $\pm$ 2.5% nationwide to 77.9 $\pm$ 1.4% at the local level.

Our results showed a huge enhancement in characterizing all individual ESA LULC CCI categories, specifically for the challenging shrubland class with the lowest PAUA of 18.5 $\pm$ 1.2 and 9.7 $\pm$ 06, respectively. One reason might be the 10-m resolution of the ESA LULC CCI product, which is more compatible with regional rather than global scale. In addition, assembling approach enables substitution that improved accuracy of the last class whose prediction is the least accurate [54].

Quan.Dis and All.Dis metrics showed a similar pattern to overall accuracy, which decreased from 0.026 to 0.016 and from 0.056 to 0.049, respectively (see Tables 9 and 10). However, KA has not notably changed. Adjusted AMI decreased much below the original map, i.e., the mapping errors did not have a random origin [43]. The adjAMI score would be highly increased in our validated map if the irrelevant classes merged. We used adjusted-AMI rather than AMI in our interpretations, given that the adjusted-AMI only considers the positives of the on-diagonal of the confusion matrix. The UA and PA for the affected

class, i.e., rangeland, were slightly improved (Table 10), emphasizing the importance of site-specific validation on the LULC preliminary map.

**Table 10.** Confusion matrix and quality indicators after merging classes over the ESRS sampling scheme.

| | | | Observation | | | |
|---|---|---|---|---|---|---|
| **Predicted** | **Trees** | **Shrubland** | **Rangeland** | **Cropland** | **Built-Up** | **UA** |
| Trees | 30 | - | - | - | - | 100% |
| Shrubland | - | 27 | 3 | - | - | 90% ± 0.1 |
| Rangeland | - | 3 | 52 | 1 | 3 | 88.1% ± 0.08 |
| Cropland | - | - | 1 | 29 | - | 96.7% ± 0.07 |
| Built-up | 1 | - | - | - | 30 | 96.8% ± 0.06 |
| PA | 96.7% ± 0.06 | 90% ± 0.09 | 92.5% ± 0.07 | 96.8% ± 0.06 | 91.8% ± 0.08 | 93.5% ± 0.04 |
| PMA | 99.4% | 96.8% | 94.1% | 98.9% | 97.8% | |
| PMAP | 96.8% | 81.8% | 82.5% | 93.5% | 88.2% | |

Note: correct classifications are on the matrix diagonal; misclassifications are off-diagonal [51], UA: user's accuracy, PA: producer's accuracy, MAP: portmanteau, PMAP: partial portmanteau. Overall accuracy: 93.5% ± 0.04; allocation disagreement: 0.049; shift: 0.007; exchange: 0.04; quantity disagreement: 0.016; average mutual information: 1.91 ± 0.16; adjusted average mutual information: 1.91; kappa: 0.915 ± 0.05.

Following the map statistical improvement, the validated map was optimized, employing different ancillary datasets, such as national road and river datasets, streams derived from DEM 10 m, as well as livestock, poultry, fishery and byproduct unit GIS-ready data sets taken from the local agricultural center. As can be seen in Figure 7, the final LULC map of the sub-basin was produced following the recently added datasets and was given official approval from local experts and authorities.

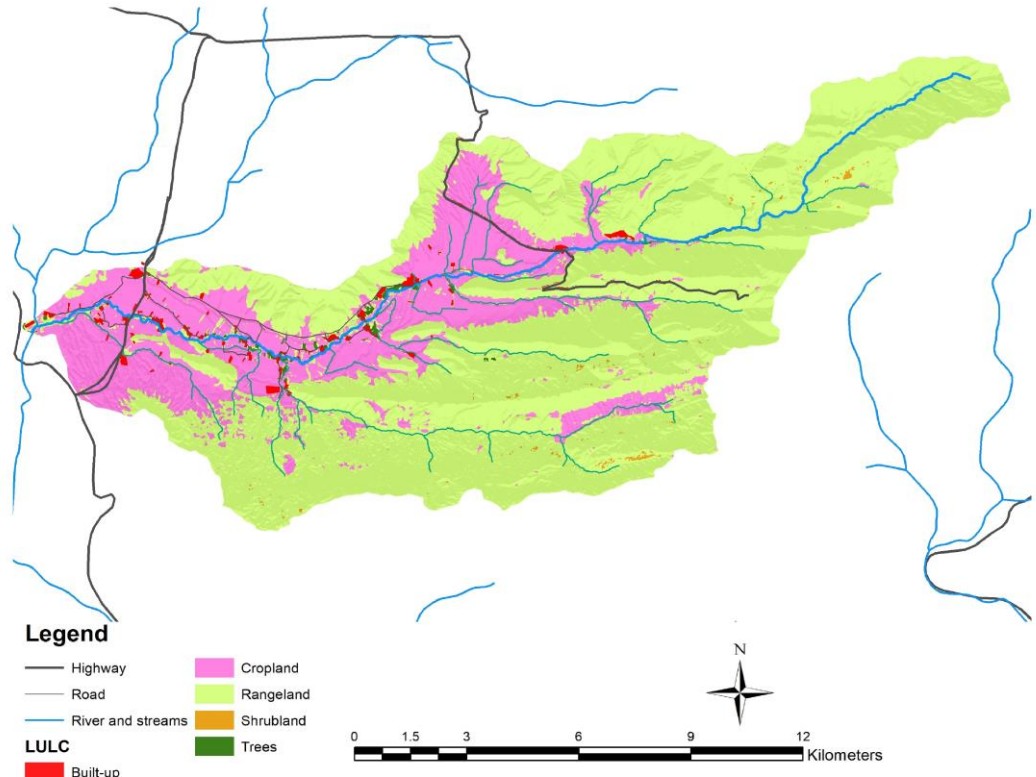

**Figure 7.** Final land use map with a spatial resolution of 10 m using the combination of reliable ancillary datasets; colors are the same as ESA LULC CCI legend, except for the new class of rangeland.

## 5. Conclusions

Since 2020, the validation of land use and land cover products has been standardized via the broad qualitative metrics LACOVAL online platform. The novel ESA LULC CCI product has not been previously assessed through diverse statistics at regional scales, specifically in mountainous drylands.

The accuracy of the ESA LULC distinguishes the area of cropland, which is the most important one with at least 73.17% accuracy for 24.2% of the whole Honam sub-basin and is reliable for application. In general, our investigation proved that well-known accuracy alone is not a good quality guideline, and the reasons why broader information-based metrics and sampling scheme should be considered specifically for class-level accuracy assessment were discussed. Actually, each accuracy metric has its own advantages and drawbacks. Therefore, map users should be aware of how the balanced approach and, in particular, the choices of LULC classes in map legend that are consistent with ground truth, impact accuracy assessment results.

The balanced ESRS worked more efficiently among sampling schemes, as it geographically distributes equal samples among different land use classes. OA is a proper metric for LULC change detection, while AUA and APA can better explain biases. We found that if the sampling scheme is appropriately selected, the calculated KA, MPA, AUA and Ave.PMAP is similar. Rangeland is a well-known land cover class in arid and semi-arid regions that is green for only 3 months and bare in other months of the year. Therefore, the barren land class of ESA LULC CCI product should be relabeled as rock/outcrop, and additionally grassland and barren land classes of the product should be merged and then relabeled as rangeland in mountainous arid and semi-arid regions. In addition, our assessment revealed that the ESA LULC CCI product greatly improved by 19.1% ± 0.06 once downscaled from the global to the regional level.

Although our findings suggest some insight into land use and land cover classification using several sampling designs at the regional scale, the limitations of a globally organized approach should be more investigated. In addition, the classes explored in this study were slightly poor. It would be interesting to investigate the ESA LULC CCI potential of classification through a similar methodology in a more diverse watershed for certain usage on regional scales. This could also be extended to full LULC classes, which provide many more possibilities for deriving landscape features. Another key point is the question of whether FROM-GLC10 and ESRI 2020 Land Cover products at such a high 10 m resolution are reliable LULC sources of data for regional scale.

Data on agricultural land cover can join water productivity data, enabling the comparison between different crop types within a region, or the same crop between different regions regarding physical, socio-economic and water resource aspects. Thus, a precise LULC base map would extend our analysis to further studies with an evaluation of crop type classification potential and the possibility of reviewing its change detection and synergic impacts. By the methodology investigated in this study, several active end users from different disciplines can provide a substantial benefit to various user communities to use well-verified products, as the primary input for all engineering aspects.

**Author Contributions:** S.M.: Data collection, conceptualization, methodology, software, validation, formal analysis, investigation, resources, data curation, writing—original draft preparation, and visualization. M.G.: Conceptualization, methodology, writing—review and editing, visualization, and supervision. M.R.B.: Conceptualization, methodology, writing—review and editing, supervision, and resources. H.A.: writing—review and editing, methodology, and supervision. N.D.: writing—review and editing, methodology, supervision, and resources. H.V.: writing—review and editing and supervision. A.M.S.—review and editing. M.C.—review and editing. All authors have read and agreed to the published version of the manuscript.

**Funding:** This research was supported by the Soil Science and Engineering Department, College of Agriculture & Natural Resources, University of Tehran, and the Soil and Water Research Institute (SWRI).

**Data Availability Statement:** The data that support the finding of this study are available from the corresponding author on reasonable request.

**Conflicts of Interest:** The authors declare no conflict of interest.

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
