# Peer review of "Deep Insight on Land Use/Land Cover Geospatial Assessment through Internet-Based Validation Tool in Upper Karkheh River Basin (KRB), South-West Iran"

_land, doi:10.3390/land12050979_

Round 1

Reviewer 1 Report

I have accepted the paper with minor suggestion please see the attachment.

Author Response

Dear reviewer,

Thank you for considering our manuscript (land-2304800) for publication in this prestigious journal. The reviewer’s comments were informative and constructive, providing a high-level guideline. We have improved our manuscript according to the suggestions which were recommended by reviewer to make it acceptable.

We have responded to all the suggestions and also made some changes to the conclusion and improved the English. We have submitted point-by-point letters with responses to the reviewers' comments. We used the track changes tool to show the changes made in the manuscript. Comments and questions were also answered as follows:

Response to reviewer Comments

I have accepted the paper with minor suggestion please see the attachment.

Comment 1: Revise the figure 1

Answer: Please specify which part should be changed.

Comment 2: Revise the figure 2

Answer: Please specify which part should be changed.

Comment 3: Paper language should checked before publication.

Answer: It has been updated.

Comment 4: Conclusion

Answer: It has been updated.

Reviewer 2 Report

The topic of the article is interesting, and appropriate for publication in LAND.

My comments for this article are as follows:

The article is very long, sometimes exhausting to read. 

Line 112 – is mentioned (Fig. 3) but is not shown in the article. It is also referred to in Line 294 (Fig. 3a), Line 334 (Fig. 3c), Line 351 (Fig. 3d) and is not in the article.

Table 10. Must be mentioned in the text.

Conclusions: The innovation of the article and the contribution of the work to the field and the progress of the research should be presented.

Author Response

Dear Reviewer,
Thank you for considering our manuscript (land-2304800) for publication in this prestigious journal. The reviewer's comments were informative and constructive and provided good guidance. We have improved our manuscript according to the suggestions recommended by the reviewers to make it acceptable.

We have considered all suggestions, including some changes to the conclusion and mention of Table 10 and Figure 3 in the text. We created point-by-point letters with responses to the editor's and reviewers' comments. We used the track change tool to show the changes made in the manuscript. Comments and questions were also responded to as follows:

The topic of the article is interesting, and appropriate for publication in LAND.

Comment 1: The article is very long, sometimes exhausting to read. 

Answer: Please indicate which parts should be shortened

Comment 2: Line 212 – is mentioned (Fig. 3) but is not shown in the article. It is also referred to in Line 294 (Fig. 3a), Line 334 (Fig. 3c), Line 351 (Fig. 3d) and is not in the article.

Answer: Figure 3 has been added.

Comment 3: Conclusions: The innovation of the article and the contribution of the work to the field and the progress of the research should be presented.

Answer: It has been updated according to reviewer’s comment.

Comment 4: Table 10. Must be mentioned in the text.

Answer: It has been added.

Reviewer 3 Report

General thoughts:

The Authors presented their research conducted in the field of land use classification using various internet platforms and IT tools. The presented research uses open Internet resources, therefore it can be considered not very revealing. However, the insight of the authors and the precise and multithreaded way of verifying the obtained results and validating the map significantly increases the assessment of the quality of the presented article.

Detailed notes:

1. Avoid including in the article title abbreviations without their expansion and proper names without their characteristics. For this reason, the proper name LACOVAL contained in the title should be expanded or eliminated. Including information about the research region (at least the name of the country) in the title of the article should also be considered.

2. All abbreviations (acronyms) should be explained in the article. Most of them have their extensions, but some of them lack full names.

3. In Chapter 2. Materials and Methods - 2.1. Study area the Authors write that they decided to assess the accuracy and usefulness of the ESA LULC CCI product on a regional scale, based on the described case study, i.e. on a regional scale (line 122-124). In order to assess the accuracy and usefulness of a product, you must have reference values against which to make such an assessment. However, the Authors, did not even provide a general way of obtaining these reference values on the basis of which the system was assessed. Such a description of the method of verification by sampling for the determination of a reference data set appears only in section 2.3.2. Spatial post-classification accuracy assessment. However, it would be good already in chapter 2.1. mention that such reference data will be obtained and used to validate the surveyed land use and land cover maps at the site of the research facility.

4. In Chapter 1. Introduction, the Authors formulated two goals they intended to achieve from their research (verses 112-114). While the first goal can be considered achieved, the question contained in goal (2) in the article is not answered. Therefore, a different form of writing the thesis of the article should be considered, so that it is consistent and consistent with the results contained in Chapter 5. Conclusions.

5. The reviewer is not an outstanding specialist in the English language, however, based on the fact that some phrases are uncomfortable to read, it can be assumed that the English language used in the reviewed article is not fully correct. Perhaps the reason for this is the huge number of acronyms contained in the text of the article. However, Native-Speaker language proofreading is recommended.

Author Response

Dear Editor,

Thank you for considering our manuscript (land-2304800) for publication in this prestigious journal. The reviewer's comments were informative and constructive and provided good guidance. We improved our manuscript according to the suggestions recommended by the reviewers to make it acceptable. We addressed all suggestions, including making some changes to the title, materials and methods, results and conclusion and improving the English language. We created point-by-point letters with responses to the editor's and reviewers' comments. We used the Track Changes tool to indicate changes made in the manuscript. Comments and questions were also responded to as follows:

Response to reviewer Comments

The Authors presented their research conducted in the field of land use classification using various internet platforms and IT tools. The presented research uses open Internet resources, therefore it can be considered not very revealing. However, the insight of the authors and the precise and multithreaded way of verifying the obtained results and validating the map significantly increases the assessment of the quality of the presented article.

Comment 1: Avoid including in the article title abbreviations without their expansion and proper names without their characteristics. For this reason, the proper name LACOVAL contained in the title should be expanded or eliminated. Including information about the research region (at least the name of the country) in the title of the article should also be considered.

Response: The title has been changed to “Deep insight on land use/land cover geospatial assessment through internet-based validation tool in upper Karkheh River Basin (KRB), west, Iran”.

Comment 2: All abbreviations (acronyms) should be explained in the article. Most of them have their extensions, but some of them lack full names.

Response: All acronyms within the text have been fully explained.

Comment 3: In Chapter 2. Materials and Methods - 2.1. Study area the Authors write that they decided to assess the accuracy and usefulness of the ESA LULC CCI product on a regional scale, based on the described case study, i.e. on a regional scale (line 122-124). In order to assess the accuracy and usefulness of a product, you must have reference values against which to make such an assessment. However, the Authors, did not even provide a general way of obtaining these reference values on the basis of which the system was assessed. Such a description of the method of verification by sampling for the determination of a reference data set appears only in section 2.3.2. Spatial post-classification accuracy assessment. However, it would be good already in chapter 2.1. mention that such reference data will be obtained and used to validate the surveyed land use and land cover maps at the site of the research facility.

Response: The sentences has been rephrased for more clarification.

Comment 4: In Chapter 1. Introduction, the Authors formulated two goals they intended to achieve from their research (verses 112-114). While the first goal can be considered achieved, the question contained in goal (2) in the article is not answered. Therefore, a different form of writing the thesis of the article should be considered, so that it is consistent and consistent with the results contained in Chapter 5. Conclusions.

Response: “In addition, our assessment revealed that ESA LULC CCI product greatly improved by 19.1% ± 0.06 once downscaled from global to regional level” has been added to the conclusion.

Comment 5: The reviewer is not an outstanding specialist in the English language, however, based on the fact that some phrases are uncomfortable to read, it can be assumed that the English language used in the reviewed article is not fully correct. Perhaps the reason for this is the huge number of acronyms contained in the text of the article. However, Native-Speaker language proofreading is recommended.

Response: It has been rechecked and the acronyms has been fully described.